# The Complete Lasso Tradeoff Diagram

**Hua Wang**    **Yachong Yang**    **Zhiqi Bu**    **Weijie J. Su**

University of Pennsylvania

`{wanghua, yachong, zbu, suw}@upenn.edu`

## Abstract

A fundamental problem in the high-dimensional regression is to understand the tradeoff between type I and type II errors or, equivalently, false discovery rate (FDR) and power in variable selection. To address this important problem, we offer the first *complete* tradeoff diagram that distinguishes all pairs of FDR and power that can be asymptotically realized by the Lasso with some choice of its penalty parameter from the remaining pairs, in a regime of linear sparsity under random designs. The tradeoff between the FDR and power characterized by our diagram holds no matter how strong the signals are. In particular, our results improve on the earlier Lasso tradeoff diagram of [21] by recognizing two simple but fundamental constraints on the pairs of FDR and power. The improvement is more substantial when the regression problem is above the Donoho–Tanner phase transition. Finally, we present extensive simulation studies to confirm the sharpness of the complete Lasso tradeoff diagram.

## 1   Introduction

Consider a data matrix $\boldsymbol{X} \in \mathbb{R}^{n \times p}$ with $p$ features and $n$ rows, and a response $\boldsymbol{y} \in \mathbb{R}^n$ from the standard linear model

$$\boldsymbol{y} = \boldsymbol{X}\boldsymbol{\beta} + \boldsymbol{z}, \tag{1.1}$$

where $\boldsymbol{z} \in \mathbb{R}^n$ is the noise term. In this paper, we study the Lasso, which, given a penalty parameter $\lambda > 0$, finds the solution to the convex optimization problem [22]:

$$\widehat{\boldsymbol{\beta}}(\lambda) = \underset{\boldsymbol{b} \in \mathbb{R}^p}{\operatorname{argmin}} \frac{1}{2} \|\boldsymbol{y} - \boldsymbol{X}\boldsymbol{b}\|_2^2 + \lambda \|\boldsymbol{b}\|_1. \tag{1.2}$$

A variable $j$ is selected by the Lasso if $\widehat{\beta}_j(\lambda) \neq 0$ and the selection is false if the variable is a noise variable in the sense that $\beta_j = 0$.

The Lasso is perhaps the most popular method in a high-dimensional setting where the number of features $p$ is very large and therefore sparse solutions are wished for. Here the sparsity, defined as $k = \#\{j : \beta_j \neq 0\}$, indicates that only $k < p$ out of the sea of all the explanatory variables are in effect and have non-zero regression coefficients. Owing to the sparsity of its solution, the Lasso is also widely recognized as a *feature selection* tool in practice.

Therefore, one particularly interesting and important question is to understand how well the Lasso performs as a feature selector. The best-case scenario is that one can find a well-tuned $\lambda$ such that the Lasso estimator can discover all the true variables and only the true variables, resulting in zero type I error and full power. This is known as the feature selection consistency. In theory, when the signals are strong enough, and when the sparsity is small enough, consistency can be guaranteed asymptotically [23]. In practice, nevertheless, the model selection consistency is usually a pipe dream: for example, even in a moderately sparse regime, consistency is impossible [21], and thus the tradeoff between the type I error and the statistical power is unavoidable. Therefore, it is much more

sensible to evaluate the performance of the Lasso not at some single "best" point (since there is no such point), but rather to focus on the entire Lasso path $\{\widehat{\boldsymbol{\beta}}(\lambda) : \lambda \in (0, \infty)\}$ and evaluate the overall tradeoff between the type I error and the statistical power. To assess the quality of each selected model $\{1 \leq j \leq p : \widehat{\beta}_j(\lambda) \neq 0\}$ at some $\lambda$, we use the false discovery proportion (FDP) and the true positive proportion (TPP) as measures. Formally, the FDP and TPP are defined as:

$$\text{FDP}(\lambda) = \frac{\#\{i : \widehat{\beta}_i(\lambda) \neq 0, \beta_i = 0\}}{\#\{i : \widehat{\beta}_i(\lambda) \neq 0\}}, \quad \text{TPP}(\lambda) = \frac{\#\{i : \widehat{\beta}_i(\lambda) \neq 0, \beta_i \neq 0\}}{\#\{i : \beta_i \neq 0\}}. \tag{1.3}$$

In this paper, we characterize the exact region in the TPP–FDP diagram where the Lasso tradeoff curves locate. A complete theoretical study of such a diagram will surely enhance our understanding of the advantages and limitations of Lasso. It can be used to theoretically guide the data analysis procedure and explain why Lasso has good empirical performance in certain scenarios.

## 1.1 Prior Art and Our Contribution

In [21], Su et al. proved that under the *linear sparsity* regime, where the ratio $n/p$ is roughly constant, it is impossible for the Lasso to achieve the feature selection consistency and a tight lower boundary of the tradeoff curves was established. Moreover, they showed that when $n/p$ is less than 1 (i.e., in the high-dimensional setting) and when the sparsity $k$ is large enough, the TPP of Lasso is always bounded away from 1, regardless of $\lambda$. This phenomenon is closely related to the Donoho–Tanner (DT) phase transition [10, 16, 15, 11, 13, 1]. The results in [21] can be visualized in the schematic plots in Figure 1. However, the dichotomy offered by their lower boundary does not give a complete picture: it is clear that the red region is unachievable, yet the lower boundary says little about the entire "Possibly Achievable" region above it. This ambiguity is resolved in this paper.

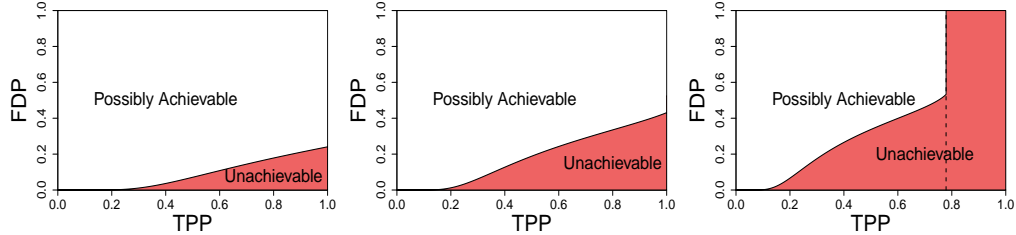

Figure 1: The dichotomy of achievability by the Lasso [21]. The sparsity ratio is fixed to $k/p = 0.3$. Left: sampling ratio $(n/p)$ is 1 (Left), 0.7 (Middle), and 0.5 (Right). The right panel depicts the case above the DT phase transition, and the dashed line represents the upper limit of TPP.

In a more recent paper [25], the authors recognized a region in the TPP–FDP diagram termed as the "Lasso Crescent" in the *noiseless* case, which is enclosed by sharp upper boundary and lower boundary. To gain more insights into those two boundaries, a new notion term as the "effect size heterogeneity" is proposed: while all the other conditions remain unchanged, the more heterogeneous the magnitudes of the effect sizes are, the better the Lasso performs. As a result, the upper boundary (or the worst tradeoff curve) in the noiseless case is given by the homogeneous effect sizes, while the lower boundary (or the best tradeoff of TPP–FDP) is achieved asymptotically by the most heterogeneous effect sizes. Though the achievability is not the primary focus in [25], they partially refine the achievable region from the entire white region in Figure 1 to the Lasso Crescent. But their scenario is limited to the case without taking the DT phase transition into account, and it is still unclear whether the entire region enclosed in the Lasso Crescent is achievable or not.

In this paper, we study the exact achievable region, taking into account the cases below and above the DT phase transition. We depict the complete Lasso achievability diagrams in terms of the TPP–FDP tradeoff in all possible scenarios. On top of [21], we specifically find three inherently different tradeoff diagrams as shown in Figure 2: We enclose the achievable region by exact boundaries, and notably, we identify two distinct sub-cases (Left and Middle panels of Figure 2) when it is below the DT phase transition, as opposed to the corresponding panels in Figure 1. Our work provides a worthwhile understanding of the DT phase transition, in the sense that we consider not only the possible region of the TPP (i.e., power), but also the possible region of (TPP, FDP) jointly.

To establish the theoretical results of our work, we leverage the powerful Approximate Message Passing (AMP) theory that has been developed recently [14, 2, 3, 12]. We also use a homotopy argument to show the asymptotic achievability of every point within the claimed region.

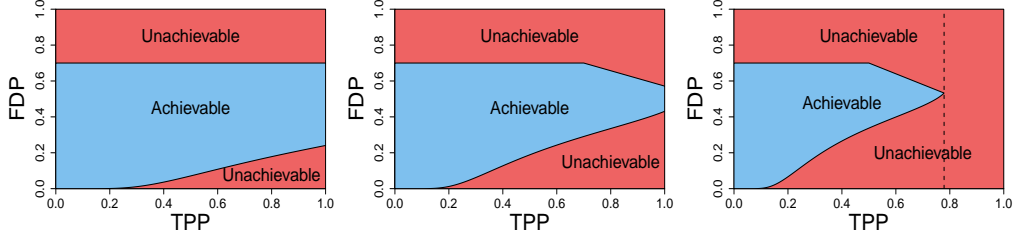

Figure 2: The complete Lasso tradeoff diagrams for the same setting as in Figure 1. The sparsity ratio is fixed to $k/p = 0.3$. The sampling ratio $n/p$ is 1 (Left), 0.7 (Middle), and 0.5 (Right). The parameters are set exactly as in Figure 1, but the diagram is now complete.

## 2  Main Results

We start presenting our results by laying out the working assumptions as follows. We assume that $X$ consists of i.i.d. $\mathcal{N}(0, 1/n)$ entries so that each column has approximately unit $\ell_2$ norm. As the index $l \to \infty$, we assume $n_l, p_l \to \infty$ with $n_l/p_l \to \delta$ in the limit for some positive constant $\delta \in (0, \infty)$. The index $l$ is often omitted for the sake of simplicity. As for the regression coefficients in the model (1.1), we assume $\beta_1, \ldots, \beta_p$ are i.i.d. copies of a random variable $\Pi$ that satisfies $\mathbb{E}\,\Pi^2 < \infty$. To model the sparse signals, we explicitly express $\Pi$ as a general mixture distribution, such that $\mathbb{P}(\Pi \neq 0) = \epsilon$ for some constant $\epsilon \in (0, 1)$, and $\Pi^\star$ is the distribution of $\Pi$ conditional on $\Pi \neq 0$, which satisfies $\mathbb{E}\,\Pi^{\star 2} < \infty$. The noise $z_i$ is i.i.d. drawn from $\mathcal{N}(0, \sigma^2)$, where $\sigma \geq 0$ is any fixed constant. For the completeness of our definition, $X, \beta$, and $z$ are jointly independent.

From now on, we consider the tradeoff between the TPP and FDP as $\lambda$ varies in $(0, \infty)$, i.e., along the Lasso path. Notation-wise, we use fdp and tpp to denote values in $[0, 1]$, and reserve TPP and FDP for the corresponding random variables.

### 2.1  The Complete Lasso tradeoff Diagram

In this subsection, we characterize the feasible region of the Lasso on the TPP–FDP diagram. We provide a set of constraints that any Lasso path must satisfy, and further show that any point in the region defined by the constraints is achievable asymptotically. We term such a region as the feasible region of Lasso.

**Definition 2.1.** For any $0 < \epsilon < 1$ and $\delta > 0$, we define the feasible region of the Lasso $\mathcal{D}_{\epsilon,\delta}$ to be the set of (tpp, fdp) pairs that satisfy the following constraints: (1) $0 \leq \text{tpp} \leq 1$; (2) $0 \leq \text{fdp} \leq 1 - \epsilon$; (3) $\text{fdp} \geq q^\star(\text{tpp})$; (4) $\frac{\epsilon}{\delta}\,\text{tpp} + \text{fdp} \leq 1$.

The function $q^\star$ in the above definition is a deterministic increasing function of $\epsilon, \delta$, which is defined in (A.2) in Appendix A and is first proposed in [21]. Recall $k$ is the sparsity $k := \#\{i : \beta_i \neq 0\}$. We now state our main theorem. To avoid any confusion, we say that a point (tpp, fdp) is asymptotically achievable, if there exist some $\Pi$, $\sigma$ and a sequence of (possibly adaptive) $\{\lambda_l\}$, such that $\|(\text{TPP}_l(\lambda_l), \text{FDP}_l(\lambda_l)) - (\text{tpp}, \text{fdp})\| \xrightarrow{\mathbb{P}} 0$.[1]

**Theorem 1** (The complete Lasso tradeoff diagram). *For any $0 < \epsilon < 1$ and $\delta > 0$, under the working assumptions, then the following holds: (a) Any Lasso tradeoff curve lies inside the region $\mathcal{D}_{\epsilon,\delta}$ asymptotically. (b) Any point in $\mathcal{D}_{\epsilon,\delta}$ is asymptotically achievable by the Lasso.*

Theorem 1 provides a complete characterization of the location of the Lasso solution on the TPP–FDP diagram. It can be seen from this theorem that the region $\mathcal{D}_{\epsilon,\delta}$ is essentially the union of *all* the Lasso

paths asymptotically:

$$\mathcal{D}_{\epsilon,\delta} = \Big\{ (t,f) : (t,f) \text{ is asymptotically achievable by some } \Pi, \sigma \text{ and } \{\lambda_l\} \Big\}.$$

We pause here to explain the intuition behind the constraints that define $\mathcal{D}_{\epsilon,\delta}$: constraint (1) directly comes from the definition; constraint (2) is from the fact that the Lasso outperforms the random guess, and hence selects no more than $1 - \epsilon$ fraction of false signals; constraint (3) is from the main result from [21, Theorem 2.1]; constraint (4) comes from the fact that the number of selected variables do not exceed $n$.

In Figure 2, we plot three different cases of the Lasso tradeoff diagrams with different $\delta$ and $\epsilon$, where the region $\mathcal{D}_{\epsilon,\delta}$ is marked in blue. The difference between these diagrams comes from different active constraints defining $\mathcal{D}_{\epsilon,\delta}$.

We rewrite the boundary lines of the constraints (2) and (4) of Definition 2.1 as

$$l_1: \quad \text{fdp} = 1 - \epsilon, \qquad l_2: \quad \text{fdp} = 1 - \frac{\epsilon}{\delta}\text{tpp}, \tag{2.1}$$

and then we can describe the three tradeoff diagrams accordingly.

Case 1: When $\delta \geq 1$ and $0 \leq \epsilon \leq 1$, $l_2$ is always above $l_1$ in the interval $(0,1)$, and thus only the constraint from $l_1$ is active. We note that this case is below the DT phase transition. This case corresponds to the left plot in Figure 2.

Case 2: When $\delta < 1$ and $\epsilon$ is sufficiently small, $l_1$ and $l_2$ intersect. Hence both constraints from $l_1$ and $l_2$ are active. However, $l_2$ is always above and thus never intersects with $q^\star$ on $(0,1)$. Similar to Case 1, it is below the DT phase transition. This case corresponds to the middle plot of Figure 2.

Case 3: When $\delta < 1$ and $\epsilon$ is sufficiently large, again $l_1$ and $l_2$ intersect, and $l_2$ also intersects with $q^\star$ within $(0,1)$. We observe the DT phase transition. This case corresponds to the right plot of Figure 2. Surprisingly, the maximum TPP achievable by the Lasso is exactly where $l_2$ and $q^\star$ intersect.

In particular, when $\delta < 1$ (i.e., Case 2 or 3), consider the following equation in $t$ [21, Equation (C.5)],

$$2(1-\epsilon)\left[ \left(1 + t^2\right)\Phi(-t) - t\phi(t) \right] + \epsilon\left(1 + t^2\right) = \delta. \tag{2.2}$$

There is a unique positive $\epsilon^\star$ such that when $\epsilon = \epsilon^\star$, (2.2) has a unique positive root in $t$. When $\epsilon > \epsilon^\star$ there is no positive root $t$. This $\epsilon^\star$ is known as the DT phase transition point. With $\epsilon^\star$, we can define the maximum achievable TPP as in [21, Lemma C.2]

$$u^\star(\delta, \epsilon) := \begin{cases} 1 - \frac{(1-\delta)(\epsilon - \epsilon^\star)}{\epsilon(1 - \epsilon^\star)}, & \delta < 1 \text{ and } \epsilon > \epsilon^\star(\delta), \\ 1, & \text{otherwise.} \end{cases} \tag{2.3}$$

When $\epsilon \leq \epsilon^\star$, Lasso can have power arbitrarily close to 1, which falls into Case 2 in our discussion above. When $\epsilon > \epsilon^\star$, the power is at most $u^\star$ and bounded away from 1, and this corresponds to Case 3. We can show in the following lemma an equivalent characterization of the DT phase transition.

**Lemma 2.2.** *The curve $q^\star$ intersects with $l_2$ in $(0,1)$ if and only if $\epsilon > \epsilon^\star(\delta)$, and in the intersecting scenario, the intersection point is at* $\text{TPP} = u^\star$.

This lemma provides an alternative view of the DT phase transition: the setting is above the DT phase transition if and only if the curve $q^\star$ and line $l_2$ intersect with each other. From this perspective, the maximum power is not a magic output of some mysterious machine, but is indeed the power when exactly $n$ discoveries are made by Lasso when it is on the lower boundary $q^\star$. The proof of this lemma can be found in Appendix A. We refer to Appendix C for more discussion on comparisons of our results with existing results.

## 3 Proofs

In this section, we establish part (a) of Theorem 1. To this end, we first show the fact that under our assumptions, the TPP and FDP have uniform limits. As will be clear in subsection 3.1, it is not hard to check that such limits of pairs (TPP, FDP) are indeed within $\mathcal{D}_{\epsilon,\delta}$.

To prove part (b) of Theorem 1, we first show that the boundary of $\mathcal{D}_{\epsilon,\delta}$ can be achieved by a sequence of priors and noises. Then, we extend the achievability result from the boundary to the entire interior of the region by a homotopy argument.

### 3.1 Proof of Part (a) of Theorem 1

We first present the following lemma to derive uniform limits of TPP and FDP over $\lambda$'s. We denote $\Pi^\star$ to be the conditional distribution of the prior given it is not zero, that is, the distribution $\Pi|\Pi \neq 0$.

**Lemma 3.1.** *(Lemma A.1 and A.2 in [21], see also Theorem 1 in [7] and Theorem 1.5 in [4]). Fix any $\delta, \epsilon, \Pi^\star, \sigma$, and $0 < \lambda_{\min} < \lambda_{\max}$. When $n/p \to \delta$ and $k/p \to \epsilon$, we have the following uniform convergence,*

$$\sup_{\lambda_{\min} < \lambda < \lambda_{\max}} |\text{FDP}(\lambda; \Pi) - \text{fdp}^\infty(\lambda; \delta, \epsilon, \Pi^\star, \sigma)| \xrightarrow{\mathbb{P}} 0, \tag{3.1}$$

$$\sup_{\lambda_{\min} < \lambda < \lambda_{\max}} |\text{TPP}(\lambda; \Pi) - \text{tpp}^\infty(\lambda; \delta, \epsilon, \Pi^\star, \sigma)| \xrightarrow{\mathbb{P}} 0, \tag{3.2}$$

*where the two deterministic functions are*

$$\text{fdp}^\infty(\lambda; \delta, \epsilon, \Pi^\star, \sigma) = \frac{2(1-\epsilon)\Phi(-\alpha)}{2(1-\epsilon)\Phi(-\alpha) + \epsilon\mathbb{P}(|\Pi^\star + \tau W| > \alpha\tau)}, \tag{3.3}$$

$$\text{tpp}^\infty(\lambda; \delta, \epsilon, \Pi^\star, \sigma) = \mathbb{P}(|\Pi^\star + \tau W| > \alpha\tau), \tag{3.4}$$

*and where $W \sim \mathcal{N}(0,1)$ independent of $\Pi$. In addition, $\tau > 0$, $\alpha > 0$ is the unique solution[2] to*

$$\tau^2 = \sigma^2 + \frac{1}{\delta}\mathbb{E}(\eta_{\alpha\tau}(\Pi + \tau W) - \Pi)^2, \quad \lambda = \left(1 - \frac{1}{\delta}\mathbb{P}(|\Pi + \tau W| > \alpha\tau)\right)\alpha\tau. \tag{3.5}$$

The guarantee of uniform convergence of TPP and FDP along the Lasso path allows us to directly deal with the two deterministic functions $\text{tpp}^\infty$ and $\text{fdp}^\infty$, instead of considering TPP and FDP as random variables for each finite $n$ and $p$, which depends on the realizations of the signal, the noise, and the design matrix. We now focus on the properties of $(\text{tpp}^\infty, \text{fdp}^\infty)$.

To prove the part (a) of Theorem 1, we prove any $(\text{tpp}^\infty, \text{fdp}^\infty)$ pair satisfies all the four constraints of Definition 2.1. The constraint (1) is trivial, since TPP is bounded between 0 and 1 by definition, and so is $\text{tpp}^\infty$. We now prove the constraints from Definition 2.1 in the order (4)(2)(3). We start by proving constraint (4), using the simple fact from the KKT condition that "Lasso never selects more than $n$ variables".

**Lemma 3.2.** *For any $\epsilon, \delta, \Pi$, and $\sigma$, we have $\frac{\epsilon}{\delta}\text{tpp}^\infty + \text{fdp}^\infty \leq 1$.*

*Proof of Lemma 3.2.* Let $V = \#\{i : \widehat{\beta}_i(\lambda) \neq 0, \beta_i = 0\}$ be the number of false discoveries, and $D = \#\{i : \widehat{\beta}_i(\lambda) \neq 0\}$ be the number of total discoveries. Recall the sparsity is $k = \#\{i : \beta_i \neq 0\}$. By definition, we have $\text{FDP} = \frac{V}{D}$ and $\text{TPP} = \frac{D-V}{k}$. So, $\frac{\epsilon}{\delta}\text{TPP}+\text{FDP} = \frac{\epsilon}{\delta}\frac{D-V}{\epsilon p} + \frac{V}{D} = \frac{D-V}{n} + \frac{V}{D} := f_{V,n}(D)$. We view $f_{V,n}(D)$ as a function of $D$, and treat $V, n$ as fixed. Note that by definition, $V \leq D$, and the number of discoveries made by the Lasso is less than or equal to $n$, so $V \leq D \leq n$ always holds. It is not hard to see that, at the two endpoints $D = n$ and $D = V$, $f_{V,n}(D)$ attains its maximum $f_{V,n}(n) = 1 = f_{V,n}(V)$. Hence, for any $V, n$, and $D$, $f_{V,n}(D) \leq 1$. By the uniform convergence of TPP and FDP in (3.1)(3.2), we finally have $\frac{\epsilon}{\delta}\text{tpp}^\infty + \text{fdp}^\infty \leq 1$. $\qquad\square$

The next lemma proves the constraint (2) in Definition 2.1 of $\mathcal{D}_{\epsilon,\delta}$.

**Lemma 3.3.** *For any $\epsilon, \delta, \Pi$, and $\sigma$, we have $\text{fdp}^\infty \leq 1 - \epsilon$.*

To see why this lemma should be intuitively true, recall that $\epsilon = k/p$, and if we select signals uniformly at random, each variable that we select is false with probability $\frac{p-k}{p} = 1 - \epsilon$. As a result of randomly selecting variables, we end up with FDP$= 1 - \epsilon$ on average. It is natural for the Lasso to produce a better result. Therefore, this lemma is a sanity check on the Lasso, claiming that the Lasso performs no worse than a random guess. Consequently, $1 - \epsilon$ serves as a simple upper bound for FDP.

*Proof on Lemma 3.3.* Observe the probability $\mathbb{P}(|\Pi^\star + \tau W| > \alpha\tau) = \mathbb{P}(|\frac{\Pi^\star}{\tau} + W| > \alpha) \geq \mathbb{P}(|W| > \alpha) = 2\Phi(-\alpha)$, where the inequality holds as the standard normal distribution is uni-modal

at the origin and $\alpha > 0$. Therefore, by (3.3) we have

$$\text{fdp}^\infty = \frac{2(1-\epsilon)\Phi(-\alpha)}{2(1-\epsilon)\Phi(-\alpha) + \epsilon\mathbb{P}(|\Pi^\star + \tau W| > \alpha\tau)} \leq \frac{2(1-\epsilon)\Phi(-\alpha)}{2(1-\epsilon)\Phi(-\alpha) + \epsilon 2\Phi(-\alpha)} \leq 1 - \epsilon. \quad \square$$

The proof of constraint (3) is involved, the goal is to show there exists some $q^*$ such that $\text{fdp}^\infty \geq q^\star(\text{tpp}^\infty)$ holds. We defer the explicit expression of $q^\star$ to (A.2), and the formal proof of this inequality to Lemma A.1 in Appendix A. Take it as given for a while, and we can prove the following:

*Proof of Theorem 1(a).* By the uniform convergence asserted in Lemma 3.1, it suffices to show the limits $(\text{tpp}^\infty, \text{fdp}^\infty)$ are in $\mathcal{D}_{\epsilon,\delta}$. Given the lemmas above, it is immediate to verify $(\text{tpp}^\infty, \text{fdp}^\infty)$ satisfies all the constraints: (1) $0 \leq \text{tpp}^\infty \leq 1$, by its definition; (2) $0 \leq \text{fdp}^\infty \leq 1 - \epsilon$, by Lemma 3.3; (3) $\text{fdp}^\infty \geq q^\star(\text{tpp}^\infty)$, by Lemma A.1; (4) $\text{fdp}^\infty + \frac{\epsilon}{\delta}\text{tpp}^\infty \leq 1$, by Lemma 3.2. $\quad \square$

## 3.2 Proof of Part (b) of Theorem 1

In the previous subsection, we have shown that, asymptotically, all (TPP, FDP) pairs along the Lasso path locate inside the region $\mathcal{D}_{\epsilon,\delta}$. We now proceed to prove that *every* point in $\mathcal{D}_{\epsilon,\delta}$ is indeed asymptotically achievable by some Lasso solution.

We note that for each prior $\Pi$, each specific noise level $\sigma$, and each fixed $\lambda$, the pair (TPP, FDP) is asymptotically a fixed point specified as the $(\text{tpp}^\infty, \text{fdp}^\infty)$ in Lemma 3.1. Therefore, given $\epsilon$ and $\delta$, the entire achievable region is composed by the trajectory of $(\text{tpp}^\infty, \text{fdp}^\infty)$ resulting from the variation of the three parameters: $\Pi, \sigma$ and $\lambda$. To simplify the analysis of the trajectory of $(\text{tpp}^\infty, \text{fdp}^\infty)$, we will always fix some of these parameters. We first "fix" the penalty parameter $\lambda$ and consider the variation of the noise and the prior. Two extreme scenarios are easy to analyze: when $\lambda$ is large enough and when $\lambda \to 0$. When $\lambda$ is large enough, the Lasso simply makes no discovery[3] for any fixed prior and noise, and hence the trajectory of $(\text{tpp}^\infty, \text{fdp}^\infty)$ is a vertical line at $\text{tpp}^\infty = 0$. When $\lambda \to 0$, there is almost zero shrinkage, so the Lasso tends to make the maximum possible amount of discoveries. It is not hard to show that when $n$ is larger than $p$, $\text{tpp}^\infty \to 1$ almost surely, while when $n$ is less than $p$, the Lasso selects at most $n$ variables. Therefore, with different sampling ratio $\delta = n/p$ and sparsity ratio $\epsilon = k/p$, one expects the pairs $(\text{tpp}^\infty, \text{fdp}^\infty)$ to have different trajectories when we vary the prior and the noise. Another useful extreme case corresponds to "fix" the prior to be some (sequence of) $\Pi$, such that when we vary $\sigma$ from 0 to $\infty$, the corresponding Lasso path is close to the lower boundary of $\mathcal{D}_{\epsilon,\delta}$ (i.e., the curve $q^\star$) and the upper boundary of $\mathcal{D}_{\epsilon,\delta}$ (i.e., $\text{fdp}^\infty = 1 - \epsilon$).

We will prove that the trajectories in the above discussion jointly constitute the boundary of $\mathcal{D}_{\epsilon,\delta}$, whose achievability is guaranteed asymptotically. Therefore, it is just a stone's throw to prove the achievability of the interior of $\mathcal{D}_{\epsilon,\delta}$ by the homotopy theory. We start with the following lemma, which analyzes the trajectory of $(\text{tpp}^\infty, \text{fdp}^\infty)$ when $\lambda \to \infty$ and $\lambda \to 0$, separately. Recall that we define $\epsilon^\star$ in equation (2.2).

**Lemma 3.4.** *For any prior $\Pi$ and any noise level $\sigma \geq 0$, let $q^\Pi$ be the corresponding TPP–FDP tradeoff curve. The following statements hold:*

*1. As $\lambda \to \infty$, we have $\lim_{\lambda\to\infty} \text{tpp}^\infty(\lambda) \to 0$, for any $\delta > 0$ and $0 < \epsilon \leq 1$;*

*2. When $\delta > 1$, as $\lambda \to 0$, we have $\lim_{\lambda\to 0}(\text{tpp}^\infty(\lambda), \text{fdp}^\infty(\lambda))$ lying on the vertical line $\text{tpp}^\infty = 1$;*

*3. When $\delta \leq 1$ and $\epsilon \geq \epsilon^\star(\delta)$, as $\lambda \to 0$, we have $\lim_{\lambda\to 0}(\text{tpp}^\infty(\lambda), \text{fdp}^\infty(\lambda))$ lying on the line $\frac{\epsilon}{\delta}\text{tpp}^\infty + \text{fdp}^\infty = 1$;*

*4. When $\delta \leq 1$ and $\epsilon < \epsilon^\star(\delta)$, as $\lambda \to 0$, we have $\lim_{\lambda\to 0}(\text{tpp}^\infty(\lambda), \text{fdp}^\infty(\lambda))$ lying on the poly-line of $\text{tpp}^\infty = 1$ and $\frac{\epsilon}{\delta}\text{tpp}^\infty + \text{fdp}^\infty = 1$.*

Next, we need to find some priors to approach the upper and lower boundary when $\sigma$ varies. We first show in the following lemma that there exists such a sequence of priors, so that when $\sigma = 0$, the Lasso tradeoff curves approximate the lower boundary $q^\star$ in (A.2).

**Lemma 3.5** (Lemma 4.4 in [25]). *Suppose $\sigma = 0$, then there exists a sequence of priors $\Pi^{(m)}$, such that as $m \to \infty$, $q^{\Pi^{(m)}}$ converges uniformly to $q^\star$.*

This lemma finds a sequence of priors that achieves the lower boundary when $\sigma = 0$. We now need to show when $\sigma \to \infty$, the tradeoff curves of the same sequence of priors approach the upper boundary $\mathrm{fdp}^\infty = 1 - \epsilon$. This is intuitive: in the presence of infinite noises, $\mathrm{fdp}^\infty$ should be $1 - \epsilon$ for any prior, since the signal to noise ratio is infinitely small, and any discovery of the Lasso is equivalent to the discovery of a random guess. Formally, we have the following lemma.

**Lemma 3.6.** *For any prior $\Pi$, when the noise level is $\sigma = \infty$, $\mathrm{fdp}^\infty \equiv 1 - \epsilon$.*

Combining Lemma 3.4, Lemma 3.5, and Lemma 3.6, it is easy to check that in those extreme cases, the $(\mathrm{tpp}^\infty, \mathrm{fdp}^\infty)$ points form the entire boundary of $\mathcal{D}_{\epsilon,\delta}$. Now we introduce a homotopy lemma to bridge from the achievability of the boundary of $\mathcal{D}_{\epsilon,\delta}$ to the achievability of the interior of $\mathcal{D}_{\epsilon,\delta}$. The idea of homotopy is pretty intuitive: suppose there are two curves, Curve $A$ and Curve $B$, and a continuous transformation $f$ move $A$ to $B$. It is easy to imagine that the trajectories of the two endpoints of Curve $A$ during the transformation $f$ form two other curves, say Curve $C$, and Curve $D$. Then there is a region surrounded by Curve $A, B, C$, and $D$. It is a region defined by Curve $A, B$, and the transformation $f$. The homotopy theory guarantees that every point in this region is passed by the transforming curve during the transformation. We formalize this idea in the following lemma. Its proof is given in Appendix B.

**Lemma 3.7** (Intermediate value theorem on the plane). *If a continuous curve in $\mathbb{R}^2$ is parameterized by $f : [l, r] \times [0, 1] \to \mathbb{R}^2$ and if the four curves: $\mathcal{C}_1 = \{f(\lambda, 0) : l \leq \lambda \leq r\}$, $\mathcal{C}_2 = \{f(\lambda, 1) : l \leq \lambda \leq r\}$, $\mathcal{C}_3 = \{f(l, t) : 0 \leq t \leq 1\}$, $\mathcal{C}_4 = \{f(r, t) : 0 \leq t \leq 1\}$, join together as a simple closed curve [4], $\mathcal{C} := \mathcal{C}_1 \cup \mathcal{C}_2 \cup \mathcal{C}_3 \cup \mathcal{C}_4$, then $\mathcal{C}$ encloses an interior area $\mathcal{D}$, and $\forall (x, y) \in \mathcal{D}$, $\exists (\lambda, t) \in \mathcal{I} \times [0, 1]$ such that $f(\lambda, t) = (x, y)$. In other words, every point inside the boundary curve $\mathcal{C}$ is realizable by some $f(\lambda, t)$.*

*Proof of Theorem 1(b).* Fix any $\delta$ and $\epsilon$. By Lemma 3.1, to prove the asymptotic achievability of $\mathcal{D}_{\epsilon,\delta}$ by $(\mathrm{TPP}, \mathrm{FDP})$, we only need to prove every point of $\mathcal{D}_{\epsilon,\delta}$ can be achieved by some $(\mathrm{tpp}^\infty, \mathrm{fdp}^\infty)$. Note that $(\mathrm{tpp}^\infty, \mathrm{fdp}^\infty)$ is a function of $\lambda, \sigma$ and $\Pi$. So we can denote $g_1 : (\lambda, \sigma, \Pi) \mapsto \mathrm{tpp}^\infty$, $g_2 : (\lambda, \sigma, \Pi) \mapsto \mathrm{fdp}^\infty$, and $g = (g_1, g_2)$. By Lemma 3.5, there exists a sequence of priors $\Pi^{(m)}$ such that the lower boundary $q^\star$ is the uniform limit of $q^{\Pi^{(m)}}$ when $\sigma = 0$. Combining this with our definition of $g$, we know that $\lim_{m \to \infty} g(\lambda, 0, \Pi^{(m)})$ is exactly the curve $(u, q^\star(u))$. By Lemma 3.6, we have $g_2(\lambda, \infty, \Pi^{(m)}) = 1 - \epsilon$ for any $m$, and thus $\lim_{m \to \infty} g_2(\lambda, \infty, \Pi^{(m)}) = 1 - \epsilon$, which is the upper boundary. We can view the lower boundary as curve $\mathcal{C}_1$ and the upper boundary as curve $\mathcal{C}_2$ in the Lemma 3.7. Given this, our goal is to find a transformation $f$ such that $f$ transforms $\mathcal{C}_1$ to $\mathcal{C}_2$ and encloses exactly the region $\mathcal{D}_{\epsilon,\delta}$. We define such a transformation $f$ as $f(\lambda, t) = \lim_{m \to \infty} g(\lambda, \tan(t \times \frac{\pi}{2}), \Pi^{(m)})$. From the uniform convergence, we know $f$ itself is continuous. It is direct to verify that $\mathcal{C}_1 = \{f(\lambda, 0) : l \leq \lambda \leq r\}$ is the lower boundary and $\mathcal{C}_2 = \{f(\lambda, 1) : l \leq \lambda \leq r\}$ is the upper boundary.

Now, consider the tradeoff curve of $q_t(\lambda) \equiv f(\lambda, t)$ where $\lambda \in [l, r]$. Notice that for each $t$, the curve $q_t$ is continuous and bounded on any $[l, r]$, and from Lemma 3.4 we know $q_t$ has a well-defined limit as $\lambda \to 0^+$, and $\lambda \to \infty$. Therefore, we can make a continuous extension of $q_t$ from $\lambda \in [l, r]$ to $\lambda \in [0, \infty]$ using the natural compactification. So, without loss of generality, we can think of $[r, l] = [0, \infty]$ as a compact interval on the extended real line. Let $\mathcal{C}_3 = \{f(0, t) : t \in [0, 1]\}$, and $\mathcal{C}_4 = \{f(\infty, t) : t \in [0, 1]\}$. Observe that the curve $\mathcal{C}_2$ corresponds to the case that $\sigma \to \infty$, or effectively $\Pi^\star \to 0$. Therefore by Lemma 3.6, it is just the line $\{(x, 1 - \epsilon) : x \in [0, x^\star]\}$, where $x^\star = \min\{1, \delta\}$ is the intersection of $\mathrm{fdp}^\infty = 1 - \epsilon$ and $\mathrm{fdp}^\infty + \frac{\epsilon}{\delta}\mathrm{tpp}^\infty = 1$. By Lemma 3.5, curve $\mathcal{C}_2$ is just the curve $q^\star$ in the range $[0, u^\star]$. By the part (1) of Lemma 3.4, $f(\infty, t)$ is always on the segment joining $(0, 0)$ and $(0, 1 - \epsilon)$, so $\mathcal{C}_3$ is just this segment. Similarly, by the part (2-4) of Lemma 3.4, $f(0, t)$ is always on the poly-line joining from $(x^\star, 1 - \epsilon)$ down to the endpoint of curve $\mathcal{C}_2(u^\star)$. We observe that $\mathcal{C} = \mathcal{C}_1 \cup \mathcal{C}_2 \cup \mathcal{C}_3 \cup \mathcal{C}_4$ is the boundary of the region $\mathcal{D}_{\epsilon,\delta}$. Therefore, every point in $\mathcal{D}_{\epsilon,\delta}$ is achievable by some $f(\lambda, t)$ by the homotopy Lemma 3.7. $\square$

## 4 Simulations

To better illustrate our theoretical results and understand the proof of Theorem 1(b), we present the following simulations where we fix the signals $\beta$ and vary the sampling ratio and the magnitude of the noise $z$. By Theorem 1, the Lasso path indeed 'swipes' the complete achievable region enclosed by the upper and lower boundaries in (2.1), and (A.2), when $\sigma$ varies from 0 to $\infty$. In the simulations, we fix $p = 1000, k = 300$, and set $n = 1000, 700$, and 500, respectively. Consequently, the sparsity ratio $\epsilon$ is fixed to 0.3 while the sampling ratio $\delta$ is $1, 0.7$, and 0.5. We note that these are the same parameters as we used in Figure 1 and Figure 2. Across all simulations, we fix $\beta$ to be a 300-sparse vector with 5 different levels of magnitudes: $\beta_j = 0.01, 0.1, 1, 10$ or 100, and each level contains $k/5 = 60$ variables. When $\sigma = 0$, the Lasso path is close to the lower boundary. As $\sigma$ increases, the Lasso path gradually moves upward and becomes worse. When $\sigma$ is sufficiently large, the Lasso tradeoff curve approaches the upper boundary and behaves similarly to the random guess. We plot the Lasso tradeoff curves for 8 levels of $\sigma$ in Figure 3. From the results, it is not hard to see the alignment of our theoretical results and the real simulations: our theoretical achievable TPP–FDP regions $\mathcal{D}_{\epsilon,\delta}$ indeed enclose all Lasso paths (up to small random errors) and the boundaries are tight. In Appendix C, we also conduct experiments with non-Gaussian or non-i.i.d. design matrix. Though it is hard to quantify the exact Lasso diagram for general design theoretically, empirically we find our diagram is *still* correct upto small differences. The R codes for the simulations and for plotting Lasso tradeoff Diagrams are available at `https://github.com/HuaWang-wharton/CompleteLassoDiagram`.

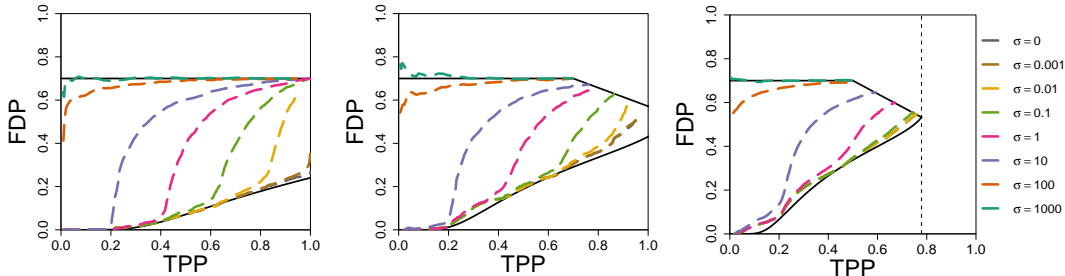

Figure 3: Simulations to show the achievability of the complete Lasso tradeoff diagrams. The lower and upper boundaries are achieved when $\sigma$ varies from 0 to 1000 with the heterogeneous prior $\beta$. We fix $p = 1000, k = 300$, and set $n = 1000$ (Left), $n = 700$ (Middle), or $n = 500$ (Right). The FDP was obtained by averaging over 100 independent trials.

## 5 Conclusions and Future Works

Our result provides the first *complete* discussion of the tradeoff between the TPP and FDP for the Lasso method under all possible circumstances. In contrast to the previous works, we focus on quantifying the exact achievable region of (TPP, FDP) pairs asymptotically, and therefore resolved the unanswered question of determining the achievability of points in the tradeoff diagram. Notably, we discover that even for the case below the DT phase transition, there is a finer sub-classification of the case $n < p$ and the case $n \geq p$ (compare the Left and Middle panels in Figure 2 to those in Figure 1). Furthermore, comparing Figure 2 to Figure 3, we confirm that our result is asymptotically exact and aligns with simulations with moderate sample size.

In closing, we introduce several directions for future research. First, it would be of interest to extend the results to other penalized regression-based models, including but not limited to the SLOPE [6], the SCAD [17], the group Lasso [18] and the sparse group Lasso [18, 20]. We believe that the homotopy tool can be applied in the search of the exact tradeoff diagrams of these methods. Such diagrams are of great theoretical interest and would surely enhance our understanding of the advantages and limitations of different methods. Second, it is interesting and important to leverage our understanding of the complete diagram to better analyze practical problems and guide more informed fine-tuning of parameters. As illustrated in Figure 4, we can have a very narrow estimate of the false discoveries when we know Lasso has large power. A complete tradeoff diagram can be used to theoretically guide our analysis and to explain why certain methods have good empirical performance in certain scenarios. This can be a good scaffold to develop better methods.

## Broader Impact

In this work, we provide a theoretical study to better understand the performance of Lasso as a model (feature) selector. We evaluate the performance of Lasso by TPP–FDP (i.e., type I error and power) tradeoff along the entire Lasso path. Such a plot is called the Lasso Diagram. Our result is the first complete analysis of the Lasso Diagram. We extend the previous unachievability results in [21] and the DT phase transition in [10] by discovering three different kinds of the Lasso Diagram,. Overall, our theoretical study provides a better understanding of the advantages and limitations of the Lasso as a model selector.

Our theoretical work has the potential to impact many other neighboring areas in the statistics and the ML community. The Lasso is one of the most famous ways to do the feature selection. Its solution is usually sparse, and thus can be used to select a subset of variables out of many possible explanatory variables. This way it has extensive applications in computational biology, genetics, medicine, statistical finance, among others. Though as a primitive statistical method, there are many modern variants outperform the Lasso itself, a thorough understanding of the Lasso is nevertheless a good starting point for a better understanding of other related and more powerful methods. Sometimes, the hardness of using Lasso as a model selector is also a good proxy for the hardness of the problem itself. To the least extent, it is also of great theoretical value to understand it thoroughly. There are a number of good works, for example [9, 23, 24, 5], that devote to understanding the advantages and limitations of the Lasso. Our new and complete analysis of the Lasso Diagram has the potential to provide new insight into the model selection problem itself along with other more complex model selection methods.

We hope our theoretical analysis can inspire more studies into other unsolved related questions. For example, similar questions also arise for other penalized regression-based model selection methods, like SLOPE or SCAD. A fine-grained analysis of the achievable region of those methods can be of great theoretical value and enhance our understanding of the pros and cons of those methods. For the practitioners, it is also very interesting and important to see how we can leverage our understanding of the Lasso diagram to better analyze practical problems and guide more informed fine-tuning in practice. A complete theoretical explanation of why certain methods have better empirical performance in certain scenarios is the scaffold to build better methods.

## Footnotes

[1]The $\text{TPP}_l$ and $\text{FDP}_l$ are the TPP and FDP calculated at a realization of design matrix $X \in \mathbb{R}^{n_l \times p_l}$, regression coefficients $\beta_i \overset{i.i.d}{\sim} \Pi$ and noise $z_i \sim \mathcal{N}(0, \sigma^2)$

[2]We note the first equation is known as the state evolution equation, and the second is the calibration equation. The notation $\eta_\alpha(\cdot)$ is the soft-thresholding operator defined as $\eta_\alpha(x) = \text{sign}(x)(|x| - t)_+$. We note that the solution $\alpha$ also satisfies $\alpha > \alpha_0$, where $\alpha_0$ is the unique root to $t$ in $(1 + t^2)\Phi(-t) - t\phi(t) = \frac{\delta}{2}$.

[3]We note that, this is the limiting regime when $\text{tpp}^\infty \to 0^+$, i.e. the moment when we are about to have an infinitesimal positive power. At this moment, the FDP can be non-zero, depending on the possibility of the first variable being a false variable. For notational convenience, we abuse the notation a little bit, and use $\lambda \to \infty$ to denote this limiting regime in the following.

[4] We actually want to specify the orientation of the curve $\mathcal{C}$ to be positively oriented, i.e., counter-clockwise oriented, as the convention for Jordan's curve. So if we assume $\mathcal{C}_1, \mathcal{C}_2, \mathcal{C}_3$ and $\mathcal{C}_4$ are all positively oriented, then $\mathcal{C}$ should be $\mathcal{C} := \mathcal{C}_1 \cup \mathcal{C}_4 \cup \mathcal{C}_2 \cup \mathcal{C}_3$. More details can be found in Appendix B.

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
