[Supplementary Material]

# Appendices

## A  Basic Lemmas

In this section, we prove Lemma 2.2 and Lemma 3.4. They are basic facts about the complete Lasso tradeoff diagram, and their proofs are based on the technical tool of the approximate message passing theory. We devote Appendix B for a detailed discussion of the homotopy theory.

First of all, we present the definition of the function $q^\star$ in constraint (3) in Definition 2.1. We introduce the lower boundary $q^\star$ of any Lasso tradeoff curve as follows: let $t^\star(u)$ be the largest solution to

$$\frac{2(1-\epsilon)\left[(1+t^2)\Phi(-t) - t\phi(t)\right] + \epsilon(1+t^2) - \delta}{\epsilon\left[(1+t^2)(1-2\Phi(-t)) + 2t\phi(t)\right]} = \frac{1-u}{1-2\Phi(-t)}, \tag{A.1}$$

where $\Phi(\cdot)$ and $\phi(\cdot)$ denote the cumulative density function and probability density function of the standard normal distribution, respectively. Then $q^\star$ is defined as

$$q^\star(u) = \frac{2(1-\epsilon)\Phi(-t^\star(u))}{2(1-\epsilon)\Phi(-t^\star(u)) + \epsilon u}. \tag{A.2}$$

With the definition of $q^\star$, we can now state the fundamental tradeoff between $\mathrm{tpp}^\infty$ and $\mathrm{fdp}^\infty$, which serves as a lower bound for FDP.

**Lemma A.1** (Theorem 2.1 in [21]). *For any $\epsilon, \delta, \Pi$, and $\sigma$, we have $\mathrm{fdp}^\infty \geq q^\star(\mathrm{tpp}^\infty)$.*

Next, we list two known facts for the reader's reference. Their proofs are essentially algebraic calculations from the state-evolution equation (The first equation in (3.5)).

**Lemma A.2.** *(Corollary 1.7 in [4]) Fix $\delta, \epsilon, \Pi$, and $\sigma$. $\alpha$, defined in equation (3.5), is an increasing function of $\lambda$, and $\lim_{\lambda \to \infty} \alpha(\lambda) = \infty$.*

**Lemma A.3.** *(Theorem 3.3 in [19]) Fix $\delta, \epsilon, \Pi$, and $\sigma$. $\tau$, defined in equation (3.5), is a continuously differentiable (positive) function of $\lambda$, and $\frac{d\tau^2}{d\lambda}$ has exactly a one-time sign change from negative to positive, and therefore is quasi-convex.*

The two lemmas above are properties of the parameters in the state evolution equation. We use them together with the following two lemmas to prove Lemma 3.4.

**Lemma A.4.** *Suppose $\epsilon > \epsilon^\star(\delta)$, where $\epsilon^\star$ is defined in (2.2). We have*

$$\inf_{t>0} \frac{2(1-\epsilon)\left[\left(1+t^2\right)\Phi(-t) - t\phi(t)\right] + \epsilon\left(1+t^2\right)}{\delta} > 1. \tag{A.3}$$

Though it is not hard to prove this fact using a pure calculus tool, we give a simple proof that leverages the definition of $\epsilon^\star$.

*Proof of Lemma A.4.* Let $h(t) = 2\left[\left(1+t^2\right)\Phi(-t) - t\phi(t)\right]$ and $g(t) = (1-\epsilon^\star)h(t) + \epsilon^\star(1+t^2)$ Notice that for any $t > 0$, we have

$$\frac{d}{dt}h(t) = 2[t\Phi(-t) - (1+t^2)\phi(t)] \leq 2[t\Phi(-t) - (1+t^2)(\frac{1}{t} - \frac{1}{t^3})\Phi(-t)] = -\frac{2}{t^3}\Phi(-t) < 0,$$

where the first inequality is due to the well-known fact $\phi(t) \geq (\frac{1}{t} - \frac{1}{t^3})\Phi(-t)$ for $t > 0$. So $h(t)$ is strictly decreasing when $t > 0$, and thus $h(t) < h(0) = 1 < 1 + t^2$ for all $t > 0$.

Now, suppose (A.3) does not hold, and thus we can find some $t$ such that

$$2(1-\epsilon)\left[\left(1+t^2\right)\Phi(-t) - t\phi(t)\right] + \epsilon\left(1+t^2\right) = \delta,$$

or equivalently,

$$(1-\epsilon)h(t) + \epsilon(1+t^2) = \delta.$$

Since $\epsilon > \epsilon^\star$, we know

$$g(t) = (1-\epsilon^\star)h(t) + \epsilon^\star(1+t^2) < \delta.$$

But $g(0) = 1 > \delta$ and $\lim_{t\to\infty} g(t) = \infty$. By the continuity of $g(t)$, we know there is a root for $g(t) = \delta$ in $(0, t)$ and $(t, \infty)$, contradicting with the definition of $\epsilon^\star$, or the fact that $g(t) = \delta$ has a unique positive root. $\qquad\square$

**Lemma A.5.** *for $\delta < 1$ and $\epsilon > \epsilon^\star(\delta)$, we have*

$$\lim_{\lambda \to 0^+} \tau > 0.$$

*Proof of Lemma A.5.* We prove by contradiction and suppose $\lim_{\lambda \to 0^+} \tau = 0$. By Lemma A.3, we know the only regime for this to hold is when $\tau$ monotonically decreases as $\lambda$ decreases. By easy calculation, we have

$$\lim_{M \to \infty} \mathbb{E}[(\eta_\alpha(M + W) - M)^2] = 1 + \alpha^2,$$

and

$$\mathbb{E}(\eta_\alpha(W)^2) = 2\left[\left(1 + t^2\right)\Phi(-t) - t\phi(t)\right].$$

We also note that for any $\Pi^\star \neq 0$, $\frac{\Pi^\star}{\tau} \to \infty$ as $\tau \to 0$. Therefore, by (A.3) in Lemma A.4, there exists some $\eta > 0$ such that

$$\inf_{\alpha > 0} \liminf_{\tau \to 0} \frac{1}{\delta}\left(\epsilon \mathbb{E}(\eta_\alpha(\frac{\Pi^\star}{\tau} + W) - \frac{\Pi^\star}{\tau}) + (1 - \epsilon)\mathbb{E}(\eta_\alpha(W)^2)\right)$$

$$= \inf_{\alpha > 0} \frac{2(1 - \epsilon)\left[\left(1 + \alpha^2\right)\Phi(-\alpha) - \alpha\phi(\alpha)\right] + \epsilon\left(1 + \alpha^2\right)}{\delta}$$

$$= 1 + \xi > 1.$$

This inequality implies that when $\tau$ is small enough, we must have

$$\frac{1}{\delta}\left(\epsilon \mathbb{E}(\eta_\alpha(\frac{\Pi^\star}{\tau} + W) - \frac{\Pi^\star}{\tau}) + (1 - \epsilon)\mathbb{E}(\eta_\alpha(W)^2)\right) > 1 + \frac{\xi}{2}.$$

For this $\tau$, from the state evolution equation (3.5), we have

$$\tau^2 = \sigma^2 + \tau^2\frac{1}{\delta}\left(\epsilon \mathbb{E}(\eta_\alpha(\frac{\Pi^\star}{\tau} + W) - \frac{\Pi^\star}{\tau}) + (1 - \epsilon)\mathbb{E}(\eta_\alpha(W)^2)\right)$$

$$> \sigma^2 + \tau^2(1 + \frac{\xi}{2})$$

$$\geq \tau^2,$$

which is clearly a contradiction. $\qquad\square$

Given the lemmas above, we can now prove the Lemma 3.4.

*Proof of Lemma 3.4.* To prove (1) in Lemma 3.4, recall the second equation in (3.5)

$$\lambda = \left(1 - \frac{1}{\delta}\mathbb{P}(|\Pi + \tau W| > \alpha\tau)\right)\alpha\tau.$$

Note that the quantity $(1 - \frac{1}{\delta}\mathbb{P}(|\Pi + \tau W| > \alpha\tau))$ is bounded between 0 and 1. So, when $\lambda \to \infty$ on the left-hand side, we must also have $\alpha\tau \to \infty$. Further, we have that $\alpha \to \infty$ as stated in Lemma A.2. By (3.4), we have

$$\lim_{\lambda \to \infty} \mathrm{tpp}^\infty = \lim_{\lambda \to \infty} \mathbb{P}(|\Pi^\star + \tau W| > \alpha\tau) = \lim_{\lambda \to \infty} \mathbb{P}\left(\left|\frac{\Pi^\star}{\tau} + W\right| > \alpha\right) = 0.$$

The last equality is due to Lemma A.3, which implies that $\lim_{\lambda \to \infty} \tau(\lambda)$ exist in $\mathbb{R}_{>0} \cup \{\infty\}$.

To prove (2), notice that when $\delta > 1$, we have

$$(1 - \frac{1}{\delta}\mathbb{P}(|\Pi + \tau W| > \alpha\tau)) \geq 1 - \frac{1}{\delta} > 0.$$

By the second equation in (3.5) again, we know that as $\lambda \to 0$ on the left-hand side, we must have

$$\lim_{\lambda \to 0^+} \alpha\tau = 0.$$

Since by definition $\Pi^\star \neq 0$, we must have

$$\lim_{\lambda \to 0^+} \mathrm{tpp}^\infty = \lim_{\lambda \to 0^+} \mathbb{P}(|\Pi^\star + \tau W| > \alpha\tau) = 1.$$

We now proceed to prove (3) and (4). Note that when $\delta < 1$, $\alpha_0$ is always positive. Recall $\alpha_0$ is the solution[5] $f(t) = \frac{\delta}{2}$, where $f(t) = (1 - t^2)\Phi(-t) - t\phi(t)$. Since $\frac{\mathrm{d}f(t)}{\mathrm{d}t} = 2t\Phi(-t) - 2\phi(t) < 0$ and $f(0) = \frac{1}{2}$, we know that the solution to $f(t) = \frac{\delta}{2} < \frac{1}{2}$ must be positive.

Now, for any $\Pi^\star \neq 0$ and $\sigma$, we consider the following two cases:

(a) When $\lim_{\lambda \to 0^+} \tau = 0$, we have:

$$\lim_{\lambda \to 0^+} \mathrm{tpp}^\infty = \lim_{\lambda \to 0^+} \mathbb{P}(|\Pi^\star + \tau W| > \alpha \tau) = 1.$$

(b) When $\lim_{\lambda \to 0^+} \tau > 0$, we can rearrange the second equation in (3.5) and get

$$\frac{\lambda}{\alpha \tau} = \left(1 - \frac{1}{\delta}\mathbb{P}(|\Pi + \tau W| > \alpha \tau)\right)$$

$$= \left(1 - \frac{1}{\delta}[\epsilon\,\mathbb{P}(|\Pi^\star + \tau W| > \alpha \tau) + (1 - \epsilon)2\Phi(-\alpha)]\right).$$

Take the limit $\lambda \to 0$, we have

$$\lim_{\lambda \to 0^+} \{\epsilon\,\mathbb{P}(|\Pi^\star + \tau W| > \alpha \tau) + (1 - \epsilon)2\Phi(-\alpha)\} = \lim_{\lambda \to 0^+} (1 - \frac{\lambda}{\alpha \tau})\delta = \delta. \qquad \text{(A.4)}$$

Combine (3.4) and (3.3), we have

$$\mathrm{fdp}^\infty = \frac{2(1 - \epsilon)\Phi(-\alpha)}{2(1 - \epsilon)\,\Phi(-\alpha) + \epsilon\mathrm{tpp}^\infty},$$

or equivalently,

$$\mathrm{fdp}^\infty + \frac{\epsilon\mathrm{tpp}^\infty}{\left(\epsilon\,\mathbb{P}(|\Pi^\star + \tau W| > \alpha \tau) + (1 - \epsilon)2\Phi(-\alpha)\right)} = 1.$$

Let $\lambda \to 0$ and by (A.4), we finally obtain $\lim_{\lambda \to 0^+}(\mathrm{fdp}^\infty + \frac{\epsilon}{\delta}\mathrm{tpp}^\infty) = 1$.

As we have shown in Lemma A.5, when $\epsilon \geq \epsilon^\star(\delta)$ only case (b) happens, and thus we have proven (3); while when $\epsilon < \epsilon^\star(\delta)$, both cases (a) and (b) are possible, and thus we have proven (4). $\qquad \square$

Next, we prove Lemma 2.2, which is a new characterization of the DT phase transition. It is useful to observe the following facts about $t^\star$, the unique positive root of equation (2.2) with $\epsilon = \epsilon^\star$.

**Lemma A.6.** *Let $t^\star$ be the unique positive root of equation (2.2) with $\epsilon = \epsilon^\star$. We have*

(1) $\frac{\phi(t^*)}{t^*} = \frac{\delta}{2(1-\epsilon^\star)}$;

(2) $\Phi(-t^*) = \frac{\delta - \epsilon^\star}{2(1-\epsilon^\star)}$;

(3) $t^\star = t^\star(u')$. *That is, $t^\star$ is also the solution to equation $(A.1)$ with $u = u'$, where $u' = 1 - \frac{(1-\delta)(\epsilon-\epsilon^\star)}{\epsilon(1-\epsilon^\star)}$.*

*Proof of Lemma A.6.* By Lemma C.2 in [21], $\delta$ and $\epsilon^\star(\delta)$ satisfies the following function form of $t^*$

$$\delta = \frac{2\phi(t^*)}{2\phi(t^*) + t^*(1 - 2\Phi(-t^*))}, \qquad \text{(A.5)}$$

$$\epsilon^\star = \frac{2\phi(t^*) - 2t^*\Phi(-t^*)}{2\phi(t^*) + t^*(1 - 2\Phi(-t^*))}. \qquad \text{(A.6)}$$

To prove (1), we observe

$$\frac{\delta}{2(1 - \epsilon^\star)} = \frac{\frac{2\phi(t^*)}{2\phi(t^*)+t^*(1-2\Phi(-t^*))}}{2\left(1 - \frac{2\phi(t^*)-2t^*\Phi(-t^*)}{2\phi(t^*)+t^*(1-2\Phi(-t^*))}\right)} = \frac{\phi(t^*)}{t^*}.$$

[5]It is defined in the footnote below Lemma 3.1

Similarly, we can prove (2) by

$$\frac{\delta - \epsilon^\star}{2(1 - \epsilon^\star)} = \frac{\frac{2\phi(t^*)}{2\phi(t^*) + t^*(1 - 2\Phi(-t^*))} - \frac{2\phi(t^*) - 2t^*\Phi(-t^*)}{2\phi(t^*) + t^*(1 - 2\Phi(-t^*))}}{2\left(1 - \frac{2\phi(t^*) - 2t^*\Phi(-t^*)}{2\phi(t^*) + t^*(1 - 2\Phi(-t^*))}\right)} = \Phi(-t^\star).$$

To prove (3), we plug in $t = t^\star$ into equation (A.1) with $u = u'$. The right-hand side of the equation becomes

$$\frac{1 - u'}{1 - 2\Phi(-t^*)} = \frac{\frac{(1-\delta)(\epsilon - \epsilon^\star)}{\epsilon(1 - \epsilon^\star)}}{1 - \frac{\delta - \epsilon^\star}{2(1 - \epsilon^\star)}} = \frac{\epsilon - \epsilon^\star}{\epsilon}.$$

So to verify $t^\star$ is the solution to the equation, we only need to prove

$$\frac{2(1 - \epsilon)\left[(1 + t^{\star 2})\Phi(-t^*) - t^*\phi(t^*)\right] + \epsilon(1 + t^{\star 2}) - \delta}{\epsilon\left[(1 + t^{\star 2})(1 - 2\Phi(-t^*)) + 2t^*\phi(t^*)\right]} = \frac{1 - u'}{1 - 2\Phi(-t^*)} = \frac{\epsilon - \epsilon^\star}{\epsilon},$$

which is equivalent to show

$$2(1 - \epsilon)\left[(1 + t^{\star 2})\Phi(-t^*) - t^*\phi(t^*)\right] + \epsilon(1 + t^{\star 2}) - \delta = (\epsilon - \epsilon^\star)\left[(1 + t^{\star 2})(1 - 2\Phi(-t^*)) + 2t^*\phi(t^*)\right],$$

or

$$2(1 - \epsilon^\star)\left[\left(1 + t^2\right)\Phi(-t) - t\phi(t)\right] + \epsilon^\star\left(1 + t^2\right) - \delta = 0.$$

This is true by the definition of $t^\star$, which is the solution to (2.2). $\qquad \square$

Given this result, we can now prove Lemma 2.2.

*Proof of Lemma 2.2.* We first observe that although $q^\star$ is originally defined on the interval $(0, 1)$, it can be extended to a function on $(0, \infty)$ with exactly the same form in (A.2). We consider this extended function and denote it still as $q^\star$ for convenience of notation. Since $q^\star$ is monotone, there is only one intersection point with $l_2$ on the $(0, \infty)$. We will verify the following fact:

$$\text{The extended curve } q^\star(u) \text{ intersects with } l_2 \text{ at } u = u' = 1 - \frac{(1 - \delta)(\epsilon - \epsilon^\star)}{\epsilon(1 - \epsilon^\star)}. \tag{A.7}$$

Suppose we are given this fact, then we know $\epsilon \geq \epsilon^\star$ implies $u' \geq 1$, so the original $q^\star$ does not intersect with $l_2$ in $(0, 1)$; and when $\epsilon < \epsilon^\star$, we have $0 < u' < 1$, so $q^\star$ intersects with $l_2$ in $(0, 1)$.

Now, to verify Fact (A.7), we use Lemma A.6 to calculate the function values of both $q^\star$ and $l_2$ at $u = u'$. The function value of $l_2$ at TPP $= u'$ is

$$1 - \frac{\epsilon}{\delta}u' = 1 - \frac{\epsilon^\star + \delta\epsilon - \epsilon\epsilon^\star - \delta\epsilon^\star}{\delta(1 - \epsilon^\star)} = \frac{(\delta - \epsilon^\star)(1 - \epsilon)}{\delta(1 - \epsilon^\star)}.$$

Since $t^\star = t^\star(u')$ by Lemma A.6, the value of $q^\star(u')$ is

$$q^\star(u') = \frac{2(1 - \epsilon)\Phi(-t^\star(u'))}{2(1 - \epsilon)\Phi(-t^\star(u')) + \epsilon u'} = \frac{2(1 - \epsilon)\frac{\delta - \epsilon^\star}{2(1 - \epsilon^\star)}}{2(1 - \epsilon)\frac{\delta - \epsilon^\star}{2(1 - \epsilon^\star)} + \epsilon(1 - \frac{(1 - \delta)(\epsilon - \epsilon^\star)}{\epsilon(1 - \epsilon^\star)})} = \frac{(\delta - \epsilon^\star)(1 - \epsilon)}{\delta(1 - \epsilon^\star)}.$$

These two quantities are equal, and therefore we know that $q^\star$ and $l_2$ intersect at $u'$. $\qquad \square$

# B  End-point Homotopy Lemmas

In this section, we will give a brief review of some basic topological concepts and facts that allow us to prove Lemma 3.7.

We first state the definition of the winding number.

**Definition B.1** (Winding number)**.** Any closed continuous curve on the $x - y$ plane that does not pass through the origin have a continuous polar form parameterization

$$r = r(t), \quad \theta = \theta(t), \quad \text{for } 0 \leq t \leq 1.$$

Since the function $r(t)$ and $\theta(t)$ are continuous and the curve starts and ends at the same point, $\theta(0)$ and $\theta(1)$ must differ by an integer multiple of $2\pi$. We define the winding number of the curve with respect to the origin by:

$$\text{winding number} = \frac{\theta(1) - \theta(0)}{2\pi}. \tag{B.1}$$

By translating the coordinate system, it extends to the definition of winding number around any point $p$.

As an easy calculus exercise, one can prove that the winding number is well-defined and generic to the curve, meaning that the winding number is the same for any parameterization. In topological terminology, the winding number of the curve with respect to the origin is also known as the degree of continuous mapping from the curve to $S^1$, the unit circle in $\mathbb{R}^2$.

Now we define the homotopic function:

**Definition B.2.** Suppose $X$ and $Y$ are topological spaces, and $f$ and $g$ are two continuous functions from $X$ to $Y$. A homotopy between $f$ and $g$ is a continuous function $H : X \times [0, 1] \to Y$, such that $H(x, 0) = f(x)$ and $H(x, 1) = g(x)$ for all $x \in X$. Such $f$ and $g$ are called to be homotopic.

It is easy to see the homotopic relation between functions is an equivalence relation. This can extend to an equivalence relation between topological spaces.

**Definition B.3.** Two topological spaces $X$ and $Y$ are homotopy equivalent, if there exist continuous maps $f : X \to Y$ and $g : Y \to X$, such that $g \circ f$ is homotopic to the identity map on $Id_X$ and $g \circ f$ homotopic to $Id_Y$.

A famous result given by [8] is the following lemma relates the homotopy equivalence and the winding number.

**Lemma B.4.** *[Degree is homotopy invariant] If $f, g : S^1 \to R^2$ are homotopic, then the degree of $f$ and $g$ are the same, in other words, the winding numbers of the curves $f(S^1)$ and $g(S^1)$ are the same.*

To prove Lemma 3.7, we need the following classical result on the topological characterization of a simple closed curve.

**Lemma B.5.** *[Jordan-Schoenflies theorem] Every simple closed curve on the plane separates the plane into two regions, one (the "inside") bounded and the other (the "outside") unbounded; further these two regions are homeomorphic (and thus homotopic) to the inside and outside of a standard circle $S^1$ on the plane. Specifically, the "inside" region is contractible, i.e., homotopy equivalent to a point.*

*Proof of Lemma 3.7.* We prove by contradiction and suppose there is some point $p \in \mathcal{D}$ that is not in the image of $f$. Denote the domain as $A = \mathcal{I} \times [0, 1]$. Consider a contraction map of the domain, that is, $g : A \to A$ such that $g(A) = \{q\}$ for some point $q \in A$. By the fact that the simply connected region $A$ is contractible, we know $g$ is homotopic to $Id_A$, the identity map on $A$. Thus, by the fact that composition of homotopic functions is still homotopic, we know $f = f \circ Id_A$ is homotopic to $f' = f \circ g$, where $f'$ maps the whole domain to one point $f(q) \in \mathcal{D}$.

Now, we consider the confined maps of $f$ and $f'$ on the boundary of domain $\partial A$. By assumption, $f(\partial A) \in \mathcal{D}$ is just the boundary curve $\mathcal{C}$, while $f \circ g(A) = \{f(q)\} \in \mathcal{D}$ is just a point in $\mathcal{D}$. Since those two continuous maps are homotopic, they have the same degree, i.e., they have an equal winding number with respect to point $p$. Note that the two winding numbers with respect to $p$ must be well-defined, since, by assumption, $p$ is not in the image of $f$, so $p \notin \mathcal{C}$ and $q \neq p$.

By Jordan-Schoenflies theorem B.5, we know the winding number of $\mathcal{C}$ with respect to $p$ is 1, since it equals to the winding number of $S^1$ with respect to the origin. On the other hand, the winding number of trivial curve $\{q\}$ with respect to $p$ is 0, since $p \neq q$. This leads to a contradiction. $\qquad\square$

# C  Additional Illustrations

In this section, we present additional discussion of our work.

We start with a more illustrative discussion on the difference between our findings in Theorem 1 with the results in [21]. We prove here that all asymptotically achievable points indeed constitute $\mathcal{D}_{\epsilon,\delta}$, while [21] only showed that all Lasso paths are above the curve $q^\star$ without further specification of the achievable region nor the unachievable one. Furthermore, when it is below the DT phase transition, we separate the Case 1 and Case 2 (the Left and Middle panels in Figure 2) which is not distinguished therein. Those two diagrams are very different. The case in the Middle panel guarantees that one cannot make too many mistakes with full power, since the FDP has a non-trivial upper bound.

To elaborate on the last point, we demonstrate three more tradeoff diagrams below, focusing on the case when $n \leq p$. As it is clear from Figure 4, our complete Lasso tradeoff diagrams show that the achievable region is relatively narrow in its vertical direction when the TPP is large (close to 1). In the left and right panel, we see that it is above the DT phase transition in both cases, and thus there is a single value of FDP (around 21% and 16%) when the TPP achieves its maximum. From the middle panel, we see that the range of the FDP is also narrow (36% ∼ 41%) when the TPP is close to 1. We want to emphasize that according to the result in [21], the lower bounds of FDP in all cases (21%, 36%, and 16%) are the best possible value achievable when the TPP is close to its maximum. However, our complete Lasso tradeoff diagram also guarantees that it is impossible to have a much worse FDP than the best possible ones when the TPP is large.

Figure 4: The complete Lasso tradeoff diagrams, in which high TPP guarantees low FDP. Left: sparsity ratio is $k/p = 0.722$, and sampling ratio is $n/p = 0.95$. The maximum FDP is around $0.21$ when the TPP is close to $1$. Middle: $k/p = 0.5$, $n/p = 0.85$. The maximum FDP is no more than $0.41$ when the TPP is close to $1$. Right: $k/p = 0.8$, $n/p = 0.85$. The maximum FDP is around $0.16$ when the TPP is close to $0.89$.

Next, we show empirically that our Lasso diagram, though proved under Gaussian assumptions, is *still* correct up to small differences on the lower boundary for a wide range of designs. For example, in Figure 5, we illustrate the Lasso diagram for various designs: namely, Gaussian design with AR(0.05) covariance matrix (Left), Bernoulli design with each entry being i.i.d. Bern(0.5) (Middle), and Cauchy design with each entry being Cauchy$(0, 1/n)$ (Right). In the Gaussian and Bernoulli case, our claimed region (enclosed by the black lines) is still almost exact. When the design comes from Cauchy distribution, where its mean or variance is not even well-defined, the simulation result has a higher lower boundary. This is easy to understand: the difficulty of the Cauchy design complicates the model selection problem, and the Lasso generally cannot achieve the best case as in the i.i.d. Gaussian case.

Lastly, we present in Figure 6 the level plot of the Lasso Tradeoff Diagram as suggested by the reviewers. In each diagram, we plot $\delta = n/p \, (x - \text{axis})$ versus $\epsilon = k/p \, (y - \text{axis})$, and fix FDP to be 0.2 (Left), 0.4 (Middle), and 0.6 (Right). The color of each point represents the largest TPP (since trivially, minimum TPP is 0) achievable (red for 0 and white for 1). We see that for large FDP, the TPP is always decrease with the sparsity ratio $\epsilon$, no matter beyond or below the DT phase transition. However, for small FDP, the maximum power first decreases with the increase of sparsity, and then increase with sparsity when above the DT phase transition. Our more refined result exactly characterizes this complication beyond DT transition. These plots, though being mathematically equivalent to Figure 2, complement to our tradeoff diagrams from a different perspective.

Figure 5: The Lasso tradeoff diagrams for non-i.i.d. or non-Gaussian designs. We fix sparsity ratio to be $k/p = 0.3$, and sampling ratio to be $n/p = 0.7$. The design matrix are: Gaussian design with AR(0.05) covariance matrix (Left), i.i.d. Bernoulli$(0.5)$ design (Middle), and i.i.d. Cauchy$(0, 1/n)$ design (Right).

Figure 6: The levelplot of Lasso tradeoff diagrams. We plot $\delta = n/p$ $(x - \text{axis})$ versus $\epsilon = k/p$ $(y - \text{axis})$, and fix FDP to be 0.2 (Left), 0.4 (Middle), and 0.6 (Right). The color of each point represents the largest TPP achievable