[Reviews · NeurIPS 2020]

Review 1

Summary and Contributions: The paper studies the variable selection properties of the Lasso in the problem of high dimensional linear regression with random design. The design is assumed Gaussian with (up to scaling factor) identity covariance matrix. The p-dimensional regression vector is assumed to be k-sparse. The asymptotic behavior of the pair (TPP, FDP) is studied, when (n/p)-> delta and k/p -> eps. The main result characterizes the set of the points in the plain that can be the limiting value of (TPP,FDP). == Post rebuttal update == I would like to thank the authors for their clear and detailed response. After having carefully read the response, I still think that the paper is marginally above the acceptance threshold (conditions are overly restrictive and the methodology used does not seem to be easily extended to more general situations, statistical insights of the obtained mathematical results are not of high importance). In addition, I am not convinced at all by the response on my remark on the randomness of k. Indeed, on the one hand, in the paper - line 20: k is the number of nonzero entries of beta - line 95: the same definition of k is provided - line 74: the components of beta are assumed iid drawn from the distribution k All these assumptions together clearly imply that k is random. On the other hand, - line 96: k/p is assumed to tend to eps - line 50 of the rebuttal: k = k_ell is not a random variable There is a clear contradiction in these statements and the authors have to clarify this point. In particular, in the proof of the main theorem, various results of [1] are quoted and used directly. In [1], the randomness of beta is assumed, eps is defined as the probability of beta_1 to be nonzero. Therefore, to have a statement of the theorem that matches with the proof, it seems unavoidable to impose the same assumptions and to highlight the randomness of k.

Strengths: 1. The main contributions of the paper are rather well explained and easy to understand. 2. The obtained result improves the state of the art (unless there are some very recent papers that I am not aware of).

Weaknesses: 1. The setting is rather contrained, since the covariance of the design is assumed to be the identity matrix. This is of little help to understand the behavior of the Lasso in the most interesting applications of the method, i.e. when the covariates are strongly correlated. 2. The proofs of the main contributions are not self-contained. They build on some results established in the message-passing literature, which are not recalled (neither in the main body nor in the supplementary material). 3. The statement of the main result would benefit from being more rigorous and more informative (see below).

Correctness: I could not check the correctness of the mathematical result. However, the numerical results provided in the paper are consistent with the presented theory. This can be considered as a confirmation of the validity of the main theorem.

Clarity: Up to some minor remarks listed below, the paper is well written.

Relation to Prior Work: The relation to prior work is well explained.

Reproducibility: Yes

Additional Feedback: Suggestions for improvement 1. In the statement of Theorem 1, please state explicitly what is the meaning of the word "asymptotically". In particular, since a sequence of random variables is considered, it should be made clear in which sense the convergence should be understood. 2. In the abstract and/or in the introduction, it should be clearly mentioned that only the case of uncorrelated Gaussian design is considered. 3. It is well known that the quality of the variable selection by the Lasso improves if a second step of thresholding small coefficients is performed (see for instance "Sup-norm convergence rate and sign concentration property of Lasso and Dantzig estimators" Lounici, EJS 2008). It would be interesting to provide a discussion on this. 4. In the main result of the paper, the obtained region (which is the most important object) is defined through a function q*. The definition of this function is hidden in the proofs appearing not earlier than on page 6. My suggestion is to provide the definition of q* in the definition 2.1. Furthermore, I suggest to add a subscript eps to q*, and also to t* (which is used for defining q*). It would be also very helpful to have a picture showing the shape of q* for different values of eps. 5. The title is not sufficiently informative. It would be better to somehow include in the title that the variable selection properties of the Lasso are investigated. Minor remarks/typos 1. Line 17: methods 2. Eq 1.2: replace "=" by "\in" , the solution is not necessarily unique 3. Line 31: it is quite strong to say that "there is no such thing". Such a sentence can be misinterpreted. I suggest to remove these words. 4. Line 34: inversely related generally means two quantities a and b are related by the relation a = const / b. Please rephrase this sentence to avoid any misunderstanding. 5. Line 96: k being a random variable, please clarify what is meant by k/p -> eps (in probability, almost surely, in distribution ?). 6. General remark: I guess that without loss of generality, it can be assumed throughout this work that sigma = 1. This is true since the pair (TPP,FDP) does not change if we replace (y,beta,sigma) by (y/sigma, beta/sigma,1). I suggest to make this remark and to use unit noise variance throughout the paper in order to ease notation.


Review 2

Summary and Contributions: The paper derives a complete Lasso trade-off diagram, i.e. it characterizes the entire region of feasible pairs of (FDP, TPP) along the Lasso path. Compared to earlier Lasso trade-off diagrams the authors establish an upper bound on feasible pairs of (FDP, TPP). A small-scale simulation study seems to confirm the theoretical predictions.

Strengths: This is an interesting theoretical result. I wonder how this results will be embraced by statisticians and ML researchers and how it will be used to assess Lasso estimates. The simulation study strongly support the theoretical results. This is impressive and encouraging.

Weaknesses: While the upper bound on the FDP is new, I wonder how relevant this upper bound is from a statistical perspective. What is the implication on an upper bound on the FDP? Does it imply that we should not assess models based on FDP but on the FDP divided by the upper bound? I wished the authors had discussed this in detail; the paper remains vague about the potential use and implication of this result. In contrast, the Donoho-Tanner phase transition seems to demarcate the more interesting boundary (as it puts an upper bound on the TPP).

Correctness: I am not familiar with the proof techique (AMP). However, since the simulation results match the theory very closely, I am confident that the derivations are correct.

Clarity: Yes.

Relation to Prior Work: Yes.

Reproducibility: Yes

Additional Feedback:


Review 3

Summary and Contributions: The paper provides diagrams of attainable false discovery proportion and true positive proportion (FDP and TPP) for the Lasso in moderate (p/n->c) dimensions.

Strengths: The results are very interesting and this was the best paper among my assignment. The paper characterizes in a very precise manner the pair (FDP,TPP) that are achievable by any Lasso, for any prior on the components of the truth. That result is of the same precise and simple nature as the Donoho-Tanner phase transition. Beyond this very nice theory, the paper provides more evidence towards the fact that one cannot hope to recover the exact support of the truth, and that we should think in terms of false negatives (or false discovery rate) instead.

Weaknesses: The assumptions of Gaussianity with identity covariance is usually restrictive, but I believe that this is warranted here to obtain such precise results. What happens to the Donoho-Tanner phase transition is already unclear for non identity covariance.

Correctness: ============================================= After rebuttal: The use of k as the sparsity is confusing. On lines 95, 96, 144, 161, beginning of line 162, and in the figures Figures, k is the sparsity. Then in lines 174-175 or the end of line 162 it is now clear that eps=k/p and k is nonrandom and possibly not even an integer,. It's difficult to make sense of the authors rebuttal regarding this (line 50 of the rebuttal) as k is not used in (3.3)-(3.4). I would guess that k/p->eps should simply not appear in the statements of the theorems (once epsilon and the prior are fixed, k/p->eps holds almost surely for the random k by the law of large numbers anyway). ============================================= Yes, as a plus, the empirical results (simulations) match very well with the obtained theory.

Clarity: Why is the value of p/n and sparsity not present in the three pictures in Figure 1? I am wondering what to make of these pictures without the values of sparsity and p/n. Please indicate these ratios (p/n and k/p or k/n) in every picture!

Relation to Prior Work: Yes, the contribution is substantial and significantly different from previous work. I haven't seen these precise diagrams before.

Reproducibility: Yes

Additional Feedback: As the parameters (FDP,TPP,k/n,p/n) are four dimensional, Figures 1 and 2 do not exactly convey well the relationship with the DT phase transition. I suggest to draw the following picture, that I believe would be helpful (and that I would be curious to see). In the Donoho-Tanner diagram, one plots n/p versus k/n. Fix three thresholds of FPP, say 0.2, 0.5 and 0.8. Add Figure 4 with three pictures, each a DT diagram plot with n/p versus k/n, and each point (n/p,k/n) in this 2-dimensional grid should be colored with the maximal TPP allowed with FPP smaller than threshold\in{0.2, 0.5, 0.8}. Hopefull this would reveal how fast the TPP decreases after the DT phase transition. The assumption that the prior Pi has finite second moment is required to use AMP. Is that an artefact of the proof, or is the diagram still valid (empirically in simulations) when the prior is, say, Cauchy? Does very large signals (with unbounded moment) allow to break out from the diagram (again empirically)?


Review 4

Summary and Contributions: The authors consider the performance of Lasso for variable selection. Specifically they consider the fraction of false discoveries (fdp) and the fraction of true positives (tpp) achieved by Lasso with some penalty weight lambda for a ratio n/p=delta and a fraction epsilon = k/p of non-zero entries in the unknown vector beta. For a generative model (Gaussian matrix transform, iid entries for beta, gaussian additive noise with variance sigma^2), they characterize the region D(epsilon, delta) spanned by (fdp,tpp) pairs in the large n limit, as lambda, sigma and the distribution of the beta_i are varied. This is done first by identifying the boundary of the region D(epsilon,delta), elaborating on earlier results that characterized parts of this boundary, characterizing other parts of this boundary, and finally using a homotopy argument to show that the interior of the boundary is also achievable.

Strengths: The results appear novel, and complement earlier theoretical understanding of Lasso performance for variable selection.

Weaknesses: Some glitches (unintroduced notation, some looseness in the proofs). Motivation for establishing such a diagram could be strengthened.

Correctness: As far as I can tell; the homotopy argument is only sketched (see below) but appears highly plausible, so a cleaned up proof should be reachable.

Clarity: Writing is generally fine.

Relation to Prior Work: Yes as far as I can tell.

Reproducibility: Yes

Additional Feedback: line 109: (2),(3) should be (2), (4) equation (3.5): notation eta_{alpha tau) has not been introduced I think. -line 169 corrollary --> corollary -line 205: are suffice --> suffice -line 256, "or effecively Pi^* \to 0": I didn't understand this. -line 260 "curve that approximating" --W curve approximating -typos also on line 286-287 -Line 451: the leftover comment [WH:Is this very rigorous...] is worrying! Since the homotopy argument seems so highly plausible, I would be inclined to believe the authors can tighten the proof so that no doubt remains...

[Author Response · NeurIPS 2020]

Many thanks to the reviewers for the insightful and constructive feedback, which has significantly improved the manuscript. We were pleased to see quite a few remarks collectively from the reviewers highlighting the novelty and strength of our paper, including our *state-of-the-art theoretical result* and *strongly supportive simulation*. Due to space constraints, instead of responding point-by-point, we address points in common with multiple reviews. All minor comments have been addressed and incorporated into a revised version of the paper.

**Assumption on i.i.d. Gaussian Design.** In general, AMP theory sheds light mostly on i.i.d. Gaussian data, and thus to quantify the same diagram for *general* covariance matrix, one may need to develop stronger AMP tools. As sharply pointed out by reviewer #3, though Gaussianity with identity covariance is restrictive, it is already unclear what will happen to the Donoho-Tanner phase transition without this assumption, so it is desirable yet extremely difficult to achieve this generality for our more refined result. Though it is hard to theoretically quantify the exact complete diagram for general design (especially the lower boundary), empirically we find our diagram is *still* correct up to small differences on the lower boundary for a wide range of designs. For example, in Figure 1, we illustrate the Lasso diagram for various designs: namely, Gaussian design with AR(0.05) covariance matrix (Top), Bernoulli design with each entry being i.i.d. Bern(0.5) (Middle), and Cauchy design with each entry being Cauchy($0, 1/n$) (Bottom). In the Gaussian and Bernoulli case, our claimed region (enclosed by the black lines) is still almost exact. When the design comes from Cauchy distribution, where its mean or variance is not even well-defined, the simulation result has a higher lower boundary. This is easy to understand: the difficulty of the Cauchy design complicates the model selection problem, and the Lasso generally cannot achieve the best case as in the i.i.d. Gaussian case.

**Figure 1**

**Statistical Implication from our Complete Diagram.** We want to re-emphasize the motivation and the statistical implication for studying the complete tradeoff diagram. First of all, our finding is a novel result that quantifies the exact complete achievable region of Lasso, and it is of great theoretical interest. Also, the usage of homotopy methods is rare in the statistics and the machine learning community, and our framework can be used to establish similar results for other methods like SLOPE, SCAD, group Lasso, etc. Secondly, the complete Lasso diagram allows us insight to the Lasso's performance. As illustrated in Figure 2, we can have a very narrow estimate of the false discoveries when the Lasso has large power. According to the lower bounds of FDP in the three cases ($21\%, 36\%$, and $16\%$) are the best possible value achievable when the TPP is close to its maximum. However, our complete Lasso tradeoff diagram also guarantees that it is impossible to have a much worse FDP than the best possible ones when the TPP is large.

**Figure 2**

**Level Plot of the Lasso Tradeoff Diagram.** To better illustrate our result, we present Figure 3 suggested by reviewer. In each diagram, we plot $\delta = n/p \, (x - \text{axis})$ versus $\epsilon = k/p \, (y - \text{axis})$, and fix FDP to be 0.2 (Top), 0.4 (Middle), and 0.6 (Bottom). The color of each point represents the largest TPP (since trivially, minimum TPP is 0) achievable (red for 0 and white for 1). We see that for large FDP, the TPP is always decrease with the sparsity ratio $\epsilon$, no matter beyond or below the DT phase transition. However, for small FDP, the maximum power first decreases with the increase of sparsity, and then increase with sparsity when above the DT phase transition. Our more refined result exactly characterizes this complication beyond DT transition. These plots, though being mathematically equivalent, complement to our tradeoff diagrams from a different perspective.

**Figure 3**

**Other Minor Details and Comments.** We have addressed all the corrections suggested by the reviewers and updated a revised version of the paper to define more clearly all notations and terminologies. We remark on some confusion as follows: 1. The $q^*$ is well defined when we fixed $\epsilon$ and $\delta$, however for notation simplicity we omit its dependence on $\epsilon$ and $\delta$ when it is clear from the context. 2. To be clear, as stated in the first assumption on page 3, we consider $n_l, p_l, k_l$ for some $l \geq 0$, and the asymptotic regime is when $n_l/p_l \to \delta$ and $k_l/p_l \to \epsilon$. Specifically, the $k_l$ here is not a random variable for any $l$. The prior $\Pi$ (where $k$ is random) is only used in (3.3) (3.4) to define $(\text{tpp}^\infty, \text{fdp}^\infty)$. 3. The finite second moment assumption of the prior is an assumption needed by AMP. We believe this is an artifact—in practice, a large second moment can be desirable, since it often results in large "effect-size heterogeneity" (a new notion proposed recently), where the Lasso's performance would be very close to the lower boundary $q^*$, which is also enclosed in our Lasso diagram.

[Meta-Review · NeurIPS 2020]

After extensive discussion and additional feedback from the authors the consensus is that it is an original contribution that sheds light on the limitations of the lasso as a feature selection procedure. One reviewer criticized the fact that the used techniques can only deal with independent covariates, though the authors may want to point out that this could be considered the possibly "easiest" case for feature selection, and the stated results highlight that already in this case the power of lasso is limited.